# Natural or Urban Campus Walks and Vitality in University Students: Exploratory Qualitative Findings from a Pilot Randomised Controlled Study

**DOI:** 10.3390/ijerph18042003

**Published:** 2021-02-19

**Authors:** Topaz Shrestha, Zelda Di Blasi, Marica Cassarino

**Affiliations:** School of Applied Psychology, University College Cork, T23 TK30 Cork, Ireland; z.diblasi@ucc.ie

**Keywords:** nature, vitality, energy, restoration, wellbeing, walking, health promotion, college students

## Abstract

Despite extensive evidence of the restorative effects of nature, the potential vitalizing effects of connecting with nature are yet understudied, particularly in higher education settings. University students face high levels of stress and anxiety, and may benefit from nature-based interventions that enhance positive states such as vitality. Using preliminary data from a pilot randomized controlled study with qualitative interviews, we explored the psychological experiences associated with a brief walk either in nature or an urban environment in a sample of 13 university students. The qualitative thematic analysis revealed that walking in nature was a more energizing and vitalizing experience than the urban walk. The nature walk was also found to have both affective and cognitive enhancing effects on participants. Our study highlights the usefulness of exploring subjective psychological experiences of interacting with nature, as well as supporting its restorative potential. Implications for further research and interventions are discussed.

## 1. Introduction

A substantial body of research supports nature’s restorative influence on our well-being [1,2,3,4,5]. More recently, a considerable amount of empirical evidence has emerged from disciplines such as environmental psychology, positive psychology, neuroscience, landscape aesthetics, and urban planning, amidst others, which documents the beneficial impact of nature on human health and happiness. This research demonstrates that even brief interactions with natural environments are correlated to positive physical and mental outcomes, including improved cognition [6,7], decreased blood pressure [8,9], and increases in mental well-being [10,11,12]. In a recent umbrella review of 20 systematic reviews and meta-analyses, Hossain et al. [3] found support for the claim that exposure to nature was associated with improvements in stress, anxiety, mood disorders, depressive symptoms, affect, cognitive and emotional functions, happiness, and overall mental well-being. This is also in line with a review by Bratman et al. [13], which proposed nature as a fundamental service to human’s wellbeing and health.

Stress Reduction Theory (SRT) [8] and Attention Restoration Theory (ART) [14], the two dominant theories which outline the psychological benefits of nature, delineate the multiple mechanisms through which contact with nature can alleviate stress and restore attentional capacities. While these frameworks have received extensive support in the literature, they conceive nature as a source of recuperation and the majority of the research in this area conceptualizes the benefits of nature in terms of restoration or arousal decreasing effects (e.g., stress reduction; relaxation). Joye and Dewitte [15] (p. 2), argue that the field of restoration studies has reached a theoretical standstill and that there is a ‘theoretical status quo’ which is dominated by SRT and ART despite some striking limitations inherent to these frameworks, thus calling for new ways to consider person–nature interactions, especially when considering psychological wellbeing [16].

Only recently has there been growing interest in the potential for nature not only to restore depleted mental resources, but also to enhance mental functioning and subjective wellbeing, especially within the field of Positive Psychology [16,17,18]. In their systematic review, McMahan and Estes [4] found, for instance, that the effects of contact with nature on positive affect were greater than on negative affect. An interesting psychological construct that may be linked to nature as a mentally enhancing experience is subjective vitality. Subjective vitality (SV) is a eudaimonic construct defined as subjective feelings of energy and vigor [19]. This positive feeling of aliveness is hypothesized to reflect an individual’s intrinsic well-being [5]. When vital, people experience a sentiment of enthusiasm and passion available to the self. Additionally, vitality constitutes energy which one can mobilize or regulate for specific actions [20]. SV is associated with positive stress response mechanisms and specific configurations of brain activation [21,22]. Furthermore, in vial states people exhibit greater coping and report better health and wellness [19].

The presence of nature has often been characterized as having an energizing and uplifting impact on human experience [10]. Although there is a dearth of systematic research on the connection between nature and vitality some authors have previously alluded to the correlation. For example, Stilgoe [23] suggested that the presence of nature, in everyday life, is crucial for avoiding devitalization and exhaustion. Greenway [24] found that 90% of participants allocated to a natural setting reported a greater experience of energy and aliveness. Additionally, in experiential studies De Young et al. [25] found that exposure to ordinary natural settings enhanced mental vitality. In perhaps one of the most extensive studies investigating the vitalizing effects of natural environments, Ryan et al. [26] found higher levels of subjective vitality when research participants were engaged in activities involving nature, whereas, participants who were exposed to scenes of buildings and urban settings experienced a decrease in vitality. Interestingly, Ryan and colleagues observed enhanced vitality among participants who were instructed to merely imagine themselves in nature. This evidence is encouraging in supporting the idea that nature can provide a sense of energy and can have an additive mental value.

Third level students are a population that may particularly benefit from nature experiences. University students are faced with a cohort of problems as they find themselves navigating through increased responsibility and competing demands while at college [18,27]. This time can be stressful and challenging and may manifest itself in the form of mental health problems [28,29]. Research has indicated that students who experience greater positive emotions are more emotionally stable and resilient in the face of academic challenges [29]. Conversely, students who demonstrate lower levels of positive emotions experience diminished focus and substandard academic performance in university [29,30]. Thus, one might argue that university students could significantly benefit from a natural environment intervention aimed at enhancing positive states such as vitality. While research indicates that nature-based interventions are beneficial for those suffering from mental health problems such as stress [31,32], few experimental studies to date have explored this effect on a university student sample [18,33,34]. In a recent systematic review, including 14 studies, Meredith and colleagues [34] concluded that “when contrasted with equal durations spent in urbanized settings, as little as 10 min of sitting or walking in a diverse array of natural settings significantly and positively impacted defined psychological and physiological markers of mental well-being for college-aged individuals” (p. 1). This finding supports the use of time spent in nature as a preventative measure for stress and mental health strain. If walking in nature can effectively enhance vitality and wellbeing, this would provide potent insights into the development of systematic interventions to promote psychological wellbeing in a student population. Walking in nature could be a simple, accessible and low-cost strategy to ameliorate mental health amongst students. Moreover, the effects of nature are habitually examined post-exposure; there is a need to understand the individual impact of nature during the interaction and shift the focus from outcomes to process [35,36]. There is a deficit of research which explicitly investigates what happens during exposure to nature and studies which aim to understand the participant’s tangible psychological experiences are deficient. Furthermore, there is a significant gap in the literature which employs qualitative methodologies to explore this process [13,36,37].

In order to address this gap, the primary objective of the present study was to qualitatively explore whether spending time in nature, compared to walking in an urban environment, would enhance psychological health in university students. We investigated the psychological experiences of a sample of university students’ during the 20–25 min walk in either a natural or an urban environment, in efforts to provide insight into the underpinnings of the nature wellbeing relationship and enhance our understanding of the process of walking in a natural or urban environments. Qualitative methodology, namely thematic analysis, was employed to address this objective. It was hypothesized that individuals who were randomly allocated to going for a walk in a natural environment would experience more positive states than those who were randomized to walking in an urban environment. Moreover, we hypothesized that individuals in the nature condition would perceive it to be a more restorative environment than the urban setting.

## 2. Materials and Methods

### 2.1. Design

The present qualitative investigation is part of a pilot randomized controlled study which compared the effect of walking in a natural vs. urban environment on a sample of university students. Participants were randomly assigned to walk in either a nature or an urban environment. A qualitative methodology was employed to explore the subjective experience of interacting with the environment during the walk. Additionally, quantitative methods were utilized to characterize the sample. While the study was designed to test quantitatively the effects of the walk on vitality, wellbeing and mood, the present study describes preliminary exploratory findings of subjective experiences for the sample recruited before the COVID-19 restrictions were introduced. The study adheres to the Consolidated Standards of Reporting Trials (CONSORT) guidelines [38]; a full CONSORT checklist is included in Appendix A and the CONSORT Flow chart is presented in Figure 1.

### 2.2. Participants

A sample of 13 university students (mean age = 24.92, SD = 3.55; 76.90% females) were recruited through convenience sampling and snowballing in University College Cork. Initially, 32 prospective participants had made contact with the experimenter; however, it was not possible to collect data for 19 of these participants due to the COVID-19 public health restrictions that were introduced during the study. No participants withdrew from the study. All participants read and signed a consent form, prior to participation, and gave informed consent as overseen by the university’s institutional review board. Exclusion criteria was outlined as being younger than 18 years of age and having any self-reported, current, ill health conditions that preclude participation in walking outdoors. Data collection began in February 2020 and was interrupted in March 2020 due to COVID-19 public health restrictions. All participants read an information sheet outlining the aims and nature of the study and signed a consent form prior to taking part. The study received ethical approval from the School of Applied Psychology Ethics Committee, University College Cork, Ireland (Code: MCP 2312201909, approved 24/02/2020).

### 2.3. Measures

The participants were asked two qualitative questions as soon as they finished the walk:How did you feel during the walk?Did anything worth mentioning happen during the course of your walk?

These questions were designed to aid the conceptualization of the individual impact of nature during the interaction and shift the focus from outcomes to process.

Measures that were assessed quantitatively, and that are here reported for descriptive purposes, included the perceived restorative potential of the walk, sociodemographic questions, and walking preferences.

The Perceived Restorativeness Scale (PRS-11) [39] was used to measure the restorative quality of both the natural and urban environment. The PRS-11 is an updated version of the 26-item PRS [40]. The PRS-11 is an 11 item self-reported questionnaire which asks participants to validate statements about the environment in question. Subjects judgments are made on a 0–10-point Likert scale, where 0 = not at all 6 = rather much and 10 = completely PRS-11 is invariant across gender and nationality [39]. The internal consistency of the scale was good (Cronbach’s α = 0.92).

Sociodemographic measures included gender, age, educational attainment, level of urbanity of the place of residence. Participants were also asked to indicate their environmental walking preferences, from 1 “Not at all” to 5 “Very much” (parks/green spaces, urban roads, shopping centers, wild nature).

### 2.4. Procedure

After providing written consent to take part, each participant met with the experimenter (TS) in one of the labs of the University Campus, where they completed the baseline assessment, including information on sociodemographic characteristics as well as measures of vitality, wellbeing and mood; these are not presented in this study due to the focus on qualitative findings. Each participant completed the assessment independently on a desktop computer. Participants were then randomly assigned to one of the two walk conditions, Group 1: Nature walk; Group 2: Urban walk. To achieve random group allocation, a researcher independent of the recruitment process (MC), used randomizer.org to create computer-generated random numbers, in blocks of 20; the allocation equivalent to each number was written on a sheet and was then placed in sealed opaque envelopes. Allocation was revealed by the experimenter (TS) by accessing and opening the next envelope in the sequence. After allocation, each participant embarked on the 20–25 min walk. The details of both walks were predetermined and were equated in total length (approximately 3.5 km). Both walks were located adjacent to the University Campus. All participants began and completed the walk at the same starting point (door outside the lab). The researcher explained the specific route to each participant prior to the walk. Additionally, each participant was given a map displaying the path of each walk. Participants in both groups walked on their own, while the experimenter followed behind and recorded any events of importance (i.e., weather conditions; encounters with other people etc.). The nature walk took place from the lab along a river walk, immersed in vegetation. On the other hand, the urban walk took place on an urban built-up road passing by a hospital, commercial buildings and residential estates, with small patches of shrubbery but no trees on the route.

After the walk, participants returned to the lab and completed the post-walk assessment, including the PRS-11 scale, and the walking preferences, as well as the qualitative questions related to the experience of the walk. Upon completion of the post-walk assessment, participants were debriefed on the study.

A summary of field notes taken by the experimenter during the walks is presented in Table 1; the table indicates the main circumstances overall for the nature and the urban walk, in terms of time of day when the walk was completed, the weather conditions, possible social encounters, and noise.

### 2.5. Data Analysis

A sample size estimation prior to the study using G*Power v.3.1.9.6 (Franz Faul, Universität Kiel, Germany) for a repeated measures ANOVA with within–between interaction indicated that a sample of 120 participants should be recruited in order to observe a small effect size *f* = 0.15, with alpha = 0.05, power = 0.90, two groups, two measurements, a correlation among repeated measures = 0.50, and nonsphericity correction ε = 1.00. Unfortunately, data collection was interrupted due to the pandemic restrictions; as a consequence, this study presents the results of preliminary qualitative data.

The psychological experiences of the walk described by the participants, post-walk, were analyzed qualitatively using thematic analysis. This analysis was conducted in accordance with the steps and procedures outlined by [41,42]. Thematic analysis forms the bedrock of qualitative analysis as it is concerned with meaning [41,43]. This methodology highlights patterns which reoccur throughout the data set and encompasses a particular perspective [41]. Thematic analysis minimally illustrates the data in rich detail. However, it usually expands on this initial step and interprets diverse aspects of the research question [44]. This methodology proves to be very flexible, allowing for a wide range of analytical options hence, it is suited to the present study [45]. There appeared to be explicit similarities in the way that participants described their experiences in nature, and it was evident that these patterns could form collective themes. In this study, thematic analysis was used as a functional method for working within a ‘participatory research paradigm’ whereby the participants were considered to be collaborators [41] (p. 97). The data was analyzed descriptively to gauge the participants’ personal experience during the walk. It is important to note that a phenomenological lens was adopted while analyzing the data. Phenomenology can be defined as an approach to research that seeks to describe the essence of a phenomenon by exploring it from the perspective of those who have experienced it [46]. The objective of phenomenology is to elucidate the meaning of this experience—both in terms of what was experienced and how it was experienced. Although the analysis was guided by a mental restoration lens, the codes and themes were data driven. Participants were assigned a numerical code (e.g., P1 = Participant 1), and extracts of responses are presented to elucidate the themes. Findings were cross referenced between researchers in efforts to generate the most robust themes.

In the initial stages of analysis, the participants’ transcripts were examined meticulously which resulted in a collection of 24 candidate themes. The analytical process highlighted the fact that some of the candidate themes were merely subthemes of a broader concept. After further refinement it became apparent that these candidate themes coalesced to form overarching themes. During the analytical process, it became evident that there was a distinct pattern in the way that the participants described their experiences in nature. These patterns permeated the entire data set and became the central themes. Each theme’s relative importance on the participant’s discourses was assessed with regard to how frequently they appeared in the participants’ responses. Additionally, findings were cross referenced between the three researchers and a conclusive decision, regarding the final themes, was made.

Quantitative analyses were conducted using IBM SPSS v.25 (IBM Corp, Armonk, NY, USA). Descriptive statistics for the whole sample and stratified by type of walk were calculated as mean (M), standard deviation (SD), median (Md), and interquartile range (IQR) for continuous variables; categorical variables were presented as frequencies and percentages. Given the small sample size, no inferential tests of hypotheses were carried out.

## 3. Results

### 3.1. Sample Characteristics

The sample for this study consisted of 13 participants, with a mean age = 24.92 (SD = 3.55), and of which 10 (76.90%) were females. Sample characteristics by type of walk are presented in Table 2. All the participants were university students having reached their third or higher level of education. Six (46.2%) of the participants were assigned to walk in the nature condition, and seven (53.8%) were in the urban walking condition. In terms of place of residence, most participants lived in urban areas, while three resided in the countryside.

Considering walking preferences, all participants reported to like walking in green areas and parks, as well as wild nature. Preferences for urban roads and shopping centers showed instead higher heterogeneity of responses. Considering perceptions of the walk taken for this study, the nature walk was rated as having a considerably higher restorative potential than the urban walk.

### 3.2. Qualitative Themes

The use of qualitative analytical methodologies provided us with a nuanced perspective on the participants’ psychological experiences. Analyzing the 13 participants’ qualitative responses led to the identification of three main themes as shown in Figure 2. The three dominant themes generated from the data were (1) “walking in nature is conducive to being more present in the moment and self-aware”; (2) “a walk in nature as a mood enhancing experience”; (3) “nature has the potential to restore energy levels.”

#### 3.2.1. Walking in Nature Is Conducive to Being More Present, Reflective and Self-Aware

Firstly, it was apparent that the nature walk had a cognitive and perceptual impact on the participants, and this is a theme which was intertwined throughout the responses. The walk in nature enabled participants to be more present in the moment, but also to become more self-aware and mindful of their inner states. For example, some of the participants explained how the nature walk had a grounding effect on them by drawing their attention towards the present moment and making them feel more conscious. Participant 1 emphasized this sentiment in their response:


*“Ultimately, when I slowed my thoughts and was more focused on the sensation of walking outside, I felt grounded, calm and in awe of the nature around me.” (P1)*



*“(I) noticed more buildings and parts of Cork which I haven’t seen before and from a new different perspective.” (P6)*


Participants also conveyed that the nature walk explicitly engaged their senses through sight and sound. Some participants exhibited this heightened awareness by emphasizing the fact that they felt more observant, alert and attentive to their surroundings while on the walk:


*“noticed the clouds blocking the sun and the wind picking up at various points throughout the walk.” (P1)*



*“I was fully aware of all the sounds surrounding me, particularly the sound of the river and birds singing.” (P6)*


Participant 11 conceptualized nature as something you can ‘tune into’ expounding the conception of nature having a cognitive and perceptual impact. They asserted that, while on the walk, they were:


*“able to tune into nature, noticing wildlife and spring flowers and hearing the sounds of the river and wind was rejuvenating.” (P11)*


The feeling of being able to ‘tune into’ nature exemplifies how one can immerse themselves in natural environments. By tuning into nature, the participant was able to be more attentive and notice the intricate details of the natural world which ultimately had a restorative influence. Being in nature led some participants to be more aware of their social surroundings, further contributing to an overall enhancing experience:


*“I tend to find that in atmospheres with more nature people are more amenable and present to acknowledge one another.” (P11)*


In contrast to the sentiment of feeling grounded and present while on the nature walk, participants in the urban condition reported feeling distracted and preoccupied. Certain facets of the urban setting were perceived as disturbances which diverted the participants’ attention. The urban walk stimulated the participants’ senses of sight and sound and engendered a heightened awareness. However, these effects were conceptualized in a more negative light and it is possible that the urban setting did not engage the participants’ attention in a pleasant way. The majority of the participants’ attention was drawn towards the traffic congestion and construction:


*“distracted by the noises of the traffic and construction. I found myself more hyper aware of my surroundings when having to pass by moving cars.” (P4)*



*“I was distracted by the car horns and construction.” (P13)*


The feeling of being distracted has negative connotations indicating that the traffic and construction may have been more of an interference. The hyperawareness was for safety reasons because the participant needed to be alert for danger when passing by cars. This is emblematic of how the urban setting caused the participant to be vigilant meaning that the walk was not necessarily a therapeutic or relaxing experience. When asked if anything worth mentioning happened on the walk participant 7 said: *“no, there was a lot of traffic and noise some building sites”* (P7). Similarly, participant 12 answered: *“Nothing of interest, there was a lot of pedestrians and traffic”* (P12) in response to the same question. These responses are very literal and the stimuli which engaged the participants’ attention were perceived as more of a disturbance or nuisance. The fact that both participants reported that these aspects of the urban setting were not of interest fortifies the conception of the urban setting being less compelling and perhaps more cognitively demanding than a natural space.

A reoccurring component of the natural walk was the potential for nature to precipitate a deep form of reflection. A plethora of the individuals explained how the interaction with nature evoked an intrinsic thought process while encouraging them to reflect and ultimately become more self-aware. In their response, participant 5 alluded to this enhanced reflective state and contended that while on the walk they felt: *“calm, thoughtful and reflective.”* (P5)

This introspective state highlights the potential for nature to elicit reflection. Moreover, while in the presence of nature participant 1 stated that they:


*“went through a variety of feelings as my thoughts drifted throughout the walk.” (P1)*


The extensive contemplation triggered a cohort of different feelings emphasizing the emotional magnitude of engaging with one’s thoughts. Participant 1 alluded to reflection when they differentiated between the internal world of the mind and the external world of the body in their response:


*“Internally, I drifted from thoughts about my day, and the things I plan to do after the study, to other mind wandering. When I wasn’t mind wandering I was present in my surroundings. Externally, I occasionally was distracted by people around me on the trail, and noticed the clouds blocking the sun.” (P1)*


This suggests that, while on the nature walk, the participant began to engage with her internal monologue. Here we see the sentiment of being present and aware of one’s inner states manifest itself. Additionally, this concept of ‘mind wandering’ frequently appeared in the participants’ responses. For example, participant 8 asserted:


*“I felt relaxed and at ease. I found my mind wandering and thinking about a variety of things in my life.” (P8)*


This abstraction of mind wandering might be emblematic of reflection and emphasizes how the nature walk provided participants with a safe environment to grapple with their thoughts and emotions.

Several of the participants’ responses encapsulated the tangible potential for the nature walk to alter pre-existing cognitive states. For example, participant 2 said:


*“I felt a change in my mindset, I felt my head clear of things I am stressed about and I felt lighter.” (P2)*


The participant conceptualized nature as a source of refuge from the stresses of daily life. Additionally, the change in mindset, which the individual experienced, was facilitated by walking in the natural setting. The participant contended that, while on the walk, they felt:

*“distracted from negative thoughts”* which led to them achieving *“a clearer mind.” (P2)*

In contrast to the urban setting, the natural environment provided a beneficial form of distraction from negative thought patterns. The outcome of the clearer mind is symbolic of how nature can provide clarity through the process of reflection.

Participant 5 provided the archetypal exemplar of the nature walk precipitating reflection in their response:


*“I was quite overwhelmed thinking about my problems, things I need to do, should I go home and skip class because I felt tired. But I came to a realization that it is not helpful and thinking like that will just make me become even more tired, so I decided to just focus on the present and focus on one task at a time.” (P5)*


This response encapsulates the central theme of engaging with nature as a means to be more present in the moment and aware of one’s inner states. Essentially, the participant was ruminating on problems in her life and the nature walk provided a safe domain to reflect and gain the cognizance that this behavior was detrimental. The participant exhibited extreme self-awareness in the recognition of her unhelpful thought patterns and subsequent decision to adopt a more productive behavior.

#### 3.2.2. A Walk in Nature as a Mood Enhancing Experience

Beyond the perceptual and cognitive dimension of walking outdoors, participants also talked about the impact of the walk on their affective state. Notably, while both groups described the walk as having a positive influence on their mood, participants in the nature walk described their experience as mood enhancing, whereas participants in the urban walk described a more subdued and mixed effect of the walk.

For example, participant 2 (nature walk) talked about feeling *“very positive, happy, optimistic and grateful.”* during the walk. Similarly, participant 8 said that the sounds of nature, such as the noise of the river, made them:

feel grateful that they *“weren’t walking in a car congested city center.” (P8)*

Furthermore, the nature walk provided the participants with some serenity while alleviating feelings of anxiety and worry. Participant 5 stated that they felt *“calm”* and *“less anxious”* following the exposure to nature. This highlights the potential for natural settings to alter pre-existing affective states. Simultaneously, participant 2 experienced this amelioration of mood. Referring to the feelings they experienced while on the nature walk, they said:


*“I felt my head clear of things I am stressed about and I felt lighter.” (P2)*


The walk alleviated the participant’s stress levels. This direct and positive impact on mood embodies the affective dimension and is indicative of the therapeutic effect of natural environments.

When considering the urban walk, participants described enjoying being outdoors and in good weather:


*“because of other reasons, I was a little anxious and preoccupied, but the fresh air and sun made me feel better.” (P9)*



*“I felt refreshed to be outdoors in nice weather.” (P3)*


However, the participant then goes on to say:


*“I was not that excited to be walking in a built-up, urban area for most of the walk.” (P3)*


The individual explicitly conveys that the urban setting is not their preferred walking environment and that this diminished their levels of excitement and enthusiasm.

In contrast to the additive impact of the nature walk, the urban walk appeared to have a predominantly subdued effect on the participants’ moods and an apathetic sentiment seems to pervade the majority of participants’ responses. When asked how they felt during the walk participant 7 (urban condition) said:


*“I felt fine, found my mind wandering.” (P7)*


The word ‘fine’ denotes indifference and could potentially indicate that the participant was emotionally detached during the urban walk. Furthermore, participant 10 fortifies the assertion that the urban walk had a pacifying affective impact on participants. In response to the question of how they felt during the walk they replied with the statement:


*“Neutral. It was nice going for a walk compared to sitting down indoors, especially when the weather was nice. However, the area I walked in was not particularly pleasant, e.g., the hospital and construction site would not be my preferred place to go for a walk.” (P10)*


The participant does mention that walking outdoors, in the nice weather, was enjoyable in comparison to being indoors. However, they subsequently go on to assert that the urban environment is not particularly pleasant nor is it their desired walking location.

#### 3.2.3. Nature Has the Potential to Restore Energy Levels

The starkest contrast between the two conditions was observed when comparing the participants’ responses under the theme of the energizing dimension. During the analytical process, it became evident that the energizing dimension was an idiosyncratic element of the nature walk. This theme proved to be very significant due to the fact that was closely linked to the other themes and seemed to be a catalyst eliciting affective, cognitive and perceptual effects within participants. Many of the participants emphasized the fact that the natural walk had an energizing or vitalizing effect on them. For example, when asked how they felt during the walk participant 2 said:


*“very energized, alive, very positive.” (P2)*


This highlights how the exposure to nature had an additive impact by enhancing the participant’s energy levels while simultaneously invoking more positively toned emotions. The sentiment of feeling ‘alive’ or vital is indicative of how interacting with nature precipitated positive feelings of energy and aliveness. Essentially, the participant’s enthusiasm exudes from this response which is suggestive of the potential for nature to restore energy levels and enhance vitality. Furthermore, this statement encompasses the reoccurring tendency for the nature walk to induce a surge in energy which subsequently has an impact on affective, cognitive of perceptual dimensions. The participant ultimately goes on to say how she felt:


*“happy, optimistic and grateful.” (P2)*


This influx of positive emotions could have been stimulated by the sudden increase in energy levels.

Conversely, the urban walk did not elicit the same revitalizing effects that the nature walk had on participants. The divergence between the two conditions, with regard to this theme, may provide insights into what natural environments can provide that other settings cannot. There was significant disparity between the participant’s energy levels across conditions. For example, participants 7 and 12 in the urban condition reported feeling:

*“fine”* (P7) and *“neutral”* (P12) during the urban walk.

The subdued sentiment is disparate from the high energy levels which participants’ in the nature condition experienced. This highlights the fact that the urban walk did not evoke the enlivening effects that the natural walk induced. In one instance, the urban walk appeared to deplete one of the participant’s energy levels:


*“I felt refreshed to be outdoors in nice weather, however I was not that excited to be walking in a built-up, urban area for most of the walk.” (P3)*


It is important to note that the participant found walking outdoors, in general, refreshing and revitalizing. However, the urban setting ultimately diminished the participant’s feelings of excitement and enthusiasm.

In their response, Participant 6 emphasizes how the energizing aspect of the nature walk coalesced with the cognitive and perceptual dimensions. The participant said:


*“I felt fresh, energized. Attentive to the area and to the people around me.” (P6)*


The theme of nature being a restorative source of energy is fortified by this response and the sentiment of feeling ‘fresh’ and ‘energized’ is symbolic of the revitalizing and restorative impact of nature. The contact with nature caused the participant to feel refreshed which subsequently led to the individual being increasingly present, observant and attentive to their surroundings. Participant 11 reinforced the theme of the energizing dimension when they explained how they felt on the walk:


*“I felt energized and able to tune into the nature around me, noticing wildlife and spring flowers and hearing the sounds of the river and wind was rejuvenating.” (P11)*


The nature walk enlivened the participant and evoked subjective feelings of vitality and vigor. Simultaneously, the augmentation in energy levels elicited an increase in attention allowing the participant to ‘tune into nature’ and ultimately be more present. Furthermore, the sounds and sights of the natural environment were described as ‘rejuvenating’ which is suggestive of the restorative process which occurs while engaging with nature.

The variance in energy levels across conditions is suggestive of the unique impact of natural environments and could be emblematic of the efficacy of walking in nature. The dynamic process, which occurs during interaction with nature, caused the majority of participants to feel more activated, alive and engaged with the world around them. The responses outlined embody the potential for natural environments to have a restorative effect by replenishing energy levels. Additionally, these statements highlight the fact that this is an idiosyncratic process whereby nature acts as the catalyst causing people to feel more energized which ultimately has a knock-on positive effect on mood and cognition. This process of vitalization is both restorative and additive and is central to the conceptualization of an individuals’ psychological experience while in the presence of nature.

## 4. Discussion

This study investigated, qualitatively, a sample of university students’ psychological experiences during a brief walk in either a natural or an urban environment, using qualitative accounts of the walk.

The themes generated from participants’ qualitative descriptions of the walk, highlighted a more vitalizing and energizing impact of walking in nature, as opposed to walking on the urban road, with effects associated with positive sensory, affective, and cognitive states. Furthermore, the responses revealed that participants not only exhibited a preference for the natural setting but found this environment more conducive to restoration and revitalization than participants who walked on the urban route did. We note that the qualitative descriptions made by the participants indicated perceptions of nature as a more vitalizing and energizing walking environment than the urban setting. Similarly, participants described positive effects of walking in nature on mood and wellbeing, and we observed that the restorative effect of nature was strongly linked to an enhanced sensory and cognitive experience. These findings emphasize how qualitative analysis can unearth certain subtleties which are often lost when employing solely quantitative methodology to answer nuanced research questions. While qualitative research on the vitalizing effects of nature is limited, our qualitative findings support the hypothesis that the presence of nature can be characterized as having an uplifting or energizing impact on human experience, in line with previous research which suggested that the presence of nature, in everyday life, is crucial for avoiding devitalization and exhaustion [10,23,26]. In their study on virtual environments, Plante et al. [47] found that exercise in a virtual outdoor environment energized participants, whereas exercise in an indoor setting elicited a relaxing effect. Greenway [24] reported that participants’ experience of being in an outdoor setting precipitated increased feelings of aliveness and energy. Similarly, Kaplan and Talbot [48] found that individuals who embarked on wilderness experiences frequently reported feeling more ‘alive’ and engaged with the world. These responses are similar to what participants in the current study felt which is indicative of, not only the vitalizing effect of nature, but the cognitive experience of being grounded in the present moment. Furthermore, vitalizing effects of nature have been observed in quantitative studies [26,49]; while in this study we were unable to conduct powered quantitative analyses on these effects, our findings add to growing evidence of the additive role of nature, which expands on the well-established paradigm that nature has the potential to reduce negative states [17,36].

Interestingly, participants in both conditions expressed that merely being outdoors was a positive experience. This may suggest an overall benefit of being outdoors, despite the environment, which is in line with previous research that has highlighted an overall vitalizing effect of outdoor experiences [25,26,50]. This research seems to suggest that being outdoors can per se be conducive to vitality, although, based on our qualitative findings, nature might add extra energizing effect, as observed for instance in Ryan et al. [26]. It is also possible that experiences of positive emotions in our sample, irrespective of the walking condition, could be due to the fact that the participants were engaged in physical activity while walking. The therapeutic and psychological effects of walking are abundant, and walking has been linked to multifarious health benefits including energy turnover and increased vitality [25,51]. From a practitioner’s perspective, this suggests that the emphasis can be on developing opportunities to encourage people to spend time outdoors in the everyday natural settings which are more widely available in modern society, and thus cost-effectively and rapidly benefit well-being in all community types.

The main strengths of the present study include the randomization process as well as the focus on a population which has received limited attention in terms of the vitalizing effects of nature. Students are a target group who could significantly benefit from cost- and time-effective interventions aimed at enhancing psychological wellbeing [18]. Lastly, the use of qualitative methods of data collection is a novel approach which helped to gain an in-depth understanding of the complexity of the nature–wellbeing relationship. RCTs often do not include qualitative components [52], and use of qualitative methodology enabled a more nuanced understanding of the data in the present study.

In order to increase our control of environmental conditions, we opted for two walks with very similar environmental and social circumstances; furthermore, participants were followed on the walk by the researcher who noted various factors which might have been of interest (e.g., weather conditions, walking pace, etc.). The fact that participants were aware that they were being observed may have had a social facilitation or Hawthore effect which could be a potential threat to ecological validity [53]. It is possible that the participants may have been more mindful and open to positive experiences, while on the walk, due to their awareness of being observed by the researcher [54]. While this is a potential limitation for our study, we felt that it was important to follow the participants to control the exposure given the naturalistic design of the study.

Our study is not without limitations. Firstly, data collection was interrupted due to the COVID-19 lockdown measures and this resulted in a notably small sample size, which impeded a powered quantitative analysis of the effects of the walk on vitality to integrate with the qualitative findings; thus, further data collection is warranted. When evaluating energy effects, it is important to recognize that outdoor contexts frequently involve higher physical activity and social contact. These characteristics can inflate the positive outcomes of being outdoors. Although we attempted to control for this (e.g., participants walking alone to limit social interaction), these are important confounding factors which warrant consideration in future studies. The overrepresentation of women in the sample is also a constraint of this study which limited our ability to control for potential gender differences. Furthermore, we selected a walk duration of 20–25 min to simulate a typical walk that students would undertake on our university campus, but the lack of clear guidelines on the optimal time needed to elicit effects might mean that our participants could have needed more time; however, studies have found that significant positive effects can occur after spending as little as 5 min in nature [55]. We also note that the walking environments were selected carefully, informed by the existing literature and the researchers’ knowledge of the area, but, due to the data collection constraints, we were limited in our choice of natural environment and it is possible that there was not a sufficient difference between the urban and nature walking conditions. Studies which compare extreme (very wild and dense) natural settings to urban settings seem to find more significant differences [56]. Future studies might consider manipulating both the time spent in an environment and utilizing multiple types of natural environments. We cannot entirely rule out other characteristics of outdoor environments that may have influenced our participants, such as the presence of fresh air or sunlight. For example, during both the urban and nature walk factors such as differences in the weather, external noise and the presence of other people were noted as extenuating factors by the experimenter, although the two walks were similar in terms of environmental or social circumstances. Nonetheless, we note that merely being outdoors in both environments had positive effects on the participants, supporting the idea that even urban forms of nature can provide psychological benefits. Lastly, while we used qualitative questions to elicit the experience of what might have happened to the participants during the walk, it is acknowledged that the process of nature interaction would benefit from a multidimensional investigation integrating in vivo measurements such as psychobiological or neuropsychological assessments.

Considering that the current pandemic has shifted our lifestyles and increased general anxiety, short walks in nature or green areas within urban environments may be a potential mechanism of enhancing vitality and subsequently wellbeing. During this uncertain time, we have seen people venturing outside and reconnecting with nature, perceiving it as a source of refuge. The findings of this study provide preliminary observational insights for the development of nature-based interventions aimed at enhancing energy levels among university students. It may be possible to design an intervention for students by promoting the use of campus green areas as a means to replenish depleted energy levels and enhance vitality. Such interventions would be a time- and cost-effective mechanism of improving psychological health and would have substantial positive implications for higher educational institutions. However, some important questions and significant gaps in research remain and further research is required to provide support for the implementation of such interventions. Moving forward, future studies should examine the human–nature relationship in the context of individual differences and interpersonal characteristics which may moderate the effects of nature, as recent studies have suggested [49]. Furthermore, research aimed at defining the particular kinds of nature which are most beneficial would inform practitioners developing nature–based interventions designed to boost wellbeing [12]. Attention should also be directed towards clarifying appropriate control groups when testing interventions. Much stronger support for the theoretical links between wellbeing and nature will be provided by intervention studies that have robust, active control groups, possibly even comparing nature interventions to other established methods (e.g., positive psychology interventions). Lastly, we argue that developing insight into the subjective meaning of nature should be central to future research. Not only can nature be defined by researchers, but individual natural environments epitomize personal meaning to individuals [38,57]. Sumner et al. [37] allude to the fact that investigating universality through individuality, can enhance our understanding of how nature can benefit our psychological health. By understanding more on the individual level, it may be possible to obtain multidimensional perspectives and gain insights into the mechanisms through which nature can support human health. Such research would support a more coherent and integrative approach to understanding the relationship between nature and wellbeing. Our study stimulates further research that incorporates interdisciplinary constructs; Positive and Environmental Psychology can coalesce to provide potent insights into the causal mechanisms underpinning the relationship between nature and wellbeing. By explicitly acknowledging and integrating these two areas of research, a more comprehensive and nuanced understanding of the role that nature plays in facilitating and enhancing human wellbeing can be attained.

## 5. Conclusions

In this study, we investigated the potential vitalizing effects of a short walk in nature, as opposed to walking in urban environments, for university students. Due to the pandemic, which interrupted data collection and subsequently resulted in a small sample size, our research is presented as an exploratory study. Differences between the two walks were observed through the qualitative descriptions, provided by participants after the walk. Ultimately, the nature walk was described as a more vitalizing, energizing, restorative, and cognitively enhancing environment than the urban walk. These findings may help to explain why people continually appear to be drawn to natural settings, and why as a collective we may want to think about the importance of protecting the natural elements that surround us and increasing people’s opportunities to access them. The relations between subjective vitality and restoration may be particularly interesting, as both appear to be associated with outdoors and nature in particular, and both pertain to mental wellbeing. Nature is habitually conceptualized as a source of recuperation and the majority of research explains the benefits of nature in terms of arousal decreasing effects [17]. However, the qualitative findings reveal that the exposure to nature elicited a positive high-energy state within participants. This is suggestive of the restorative value of nature as a vehicle to enhance vitality. The current study calls attention to the perception of restoration in terms of increased energy whereby spending time in nature is an ‘additive’ experience in the direction of optimal human functioning [16]. Despite the ever-expanding base of empirical evidence exploring the health benefits of nature, it is evident that qualitative methodologies are underutilized [36,37]. There is the need to consider the personal meaning that individuals hold for nature and understand the aspects of nature which are considered to be desirable in ways that are sensitive to the individual. Importantly, our study stimulates research that focuses on university students, who experience considerable mental pressures and would thus benefit from cost-effective nature-based solutions which can be incorporated into their daily routines.

## Figures and Tables

**Figure 1 ijerph-18-02003-f001:**
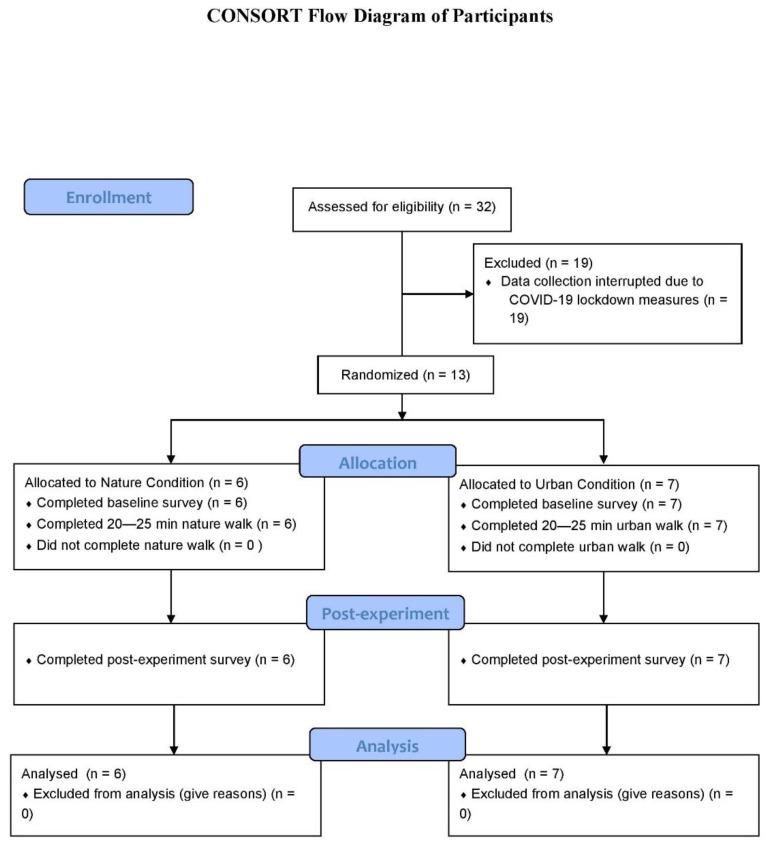
Consort flow diagram of participants.

**Figure 2 ijerph-18-02003-f002:**
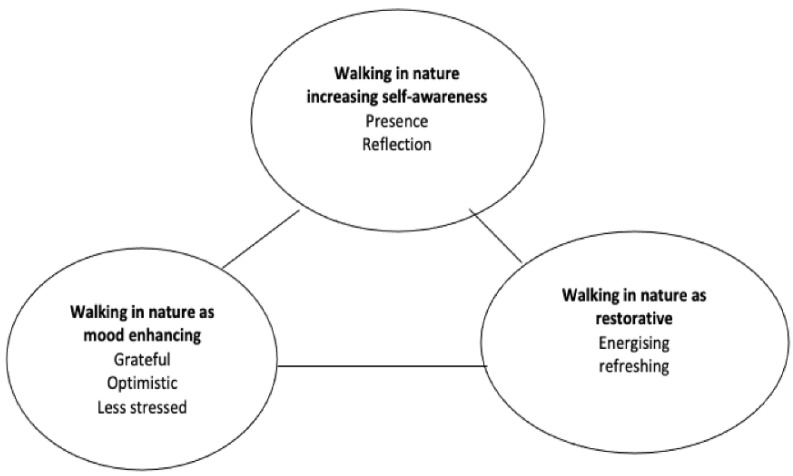
Schematic representation of the thematic findings.

**Table 1 ijerph-18-02003-t001:** Summary of field notes.

	Walk	
Dimension	Nature	Urban
Time of day	13:00–16:00 p.m.	13.00–16.00 p.m.
Weather	Sunny, blue skies for majority of participants	Good weather. Two participants experienced clouds, rain
Social encounters	Pedestrians, cyclists and animals	Higher density of pedestrians
Noise/External Stimuli	Birdsong, sounds of river and people/animals	Traffic congestion, construction

**Table 2 ijerph-18-02003-t002:** Sample characteristics by walk.

Dimension	Nature Walk	Urban Walk
Gender, n (%)		
Male	2 (33.3)	1 (14.3)
Female	3 (66.7)	6 (85.7)
Age, Mean ± SD	24.67 ± 3.33	25.14 ± 3.97
Education, n (%) Third level	6 (100)	6 (100)
Area of residence, n (%)		
Inner city	2 (33.3)	2 (28.6)
City suburb	2 (33.3)	3 (42.9)
Town	0	1 (14.3)
Countryside	2 (33.3)	1 (14.3)
Walking preference, Md (IQR)		
Green spaces	5.0 (1.0)	5.0 (1.0)
Urban roads	3.0 (2.0)	3.0 (2.0)
Shopping centers	1.5 (2.0)	2.0 (2.0)
Wild nature	5.0 (0)	5.0 (0)
Perceived restorative potential, Mean ± SD	72.6 ± 14.2	36.4 ± 11.6

IQR = Interquartile Range.

## Data Availability

Data for this study is available upon request from the study corresponding authors.

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
