# Peer review of "Natural or Urban Campus Walks and Vitality in University Students: Exploratory Qualitative Findings from a Pilot Randomised Controlled Study"

_ijerph, 2021, doi:10.3390/ijerph18042003_

Round 1
Reviewer 1 Report
Overall I thought the paper was very well presented and the writing style was easy to follow.
I appreciate the authors position that their original plans to recruit a much larger sample were not possible due to the pandemic. However I am concerned that the sample used is particularly small.
I would recommend that the authors take time to gather further data in order to allow for more robust reporting of the quantitative findings. Comparing 2 groups of 6 and 7 participants respectively is insufficient for such an analysis and publication.
In addition, the authors do not provide any detail concerning the approach taken with the thematic analysis. Were findings cross referenced between researchers for example?
On this basis I would not consider the paper for publication in it's current form. For publication to be considered additional data needs to be collected for the quantitative element and further detail needs to be given re the approach taken to the thematic analysis.
Author Response
Response to Reviewer 1 comments
Point 1: Overall I thought the paper was very well presented and the writing style was easy to follow.
Response 1: We would like to extend our sincere thanks to you for taking the time to read and review our manuscript. We are very grateful to you for providing your expertise and insight regarding our research study.
Point 2: I appreciate the authors position that their original plans to recruit a much larger sample were not possible due to the pandemic. However, I am concerned that the sample used is particularly small. I would recommend that the authors take time to gather further data in order to allow for more robust reporting of the quantitative findings. Comparing 2 groups of 6 and 7 participants respectively is insufficient for such an analysis and publication.
Response 2: We agree with the Reviewer that the sample size is small and that a larger sample size would enhance the robustness of our analysis. However, we are not in a position to continue data collection for the foreseeable future due to the pandemic restrictions. Based on the Reviewers’ comments, we have decided to present our paper as an exploratory analysis, mainly focused on our qualitative analysis. Our quantitative findings are used only to complement/support the qualitative analysis. The small sample size and preliminary nature of results have been addressed in the edited title (Natural or Urban Campus Walks and Vitality in University Students: Exploratory findings from a pilot randomised controlled study with a nested qualitative element), as well as the study objectives (lines 166-184) and methods section (358-364). In the Results section, we have focused on the qualitative findings and presented the quantitative analysis as preliminary findings, and the Discussion section has been updated accordingly to reflect these changes. See lines 435-441 and 727-745 (Results section), lines 746-770, 824-849 (Discussion Section) and 1094-1132 (Conclusion).
Point 3: In addition, the authors do not provide any detail concerning the approach taken with the thematic analysis. Were findings cross referenced between researchers for example?
Response 3: We have provided a detailed explanation of our approach taken with the qualitative analysis. Whereby, we justify our choice of thematic analysis and highlight the frameworks we followed while conducting the analysis. See lines 366-398 (Data Analysis).
Point 4: On this basis I would not consider the paper for publication in it's current form. For publication to be considered additional data needs to be collected for the quantitative element and further detail needs to be given re the approach taken to the thematic analysis.
Response 4: We hope that the revisions applied to the paper and what we have described in response to comments above will find the favour of this Reviewer. We are in complete agreement with the need for further data collection to enhance the quantitative analysis, and we are very sorry that this is not possible in the current circumstances. We believe that there is an overall benefit to publishing our work as an exploratory study focused on the qualitative analysis due to the fact that our qualitative results provide some advance towards the current knowledge on the restorative effects of nature and about the potential vitalizing effects of connecting with nature.

Reviewer 2 Report
Dear Authors
About the Originality/Novelty of the article:
- The empirical/experimental study of the restorative effects of nature, the potential vitalizing effects of connecting with nature, should be considered as a very relevant subject, with growing importance in our modern “urban” societies. The development of such a study in a higher education setting sounds also relevant, because students face high levels of stress and anxiety. The subject sounds even more relevant at pandemic restrictions context.
- It is stated at the abstract that (lines 13-14) they aimed to “compared the effectiveness of a brief walk either in nature or in an urban environment, on subjective vitality, wellbeing, mood and perceived restorative potential” in university students, and, in the introduction, they identified a primary objective (“compare the effectiveness of a 20-25 minute walk in either a natural environment or in an urban environment with heavy traffic on subjective vitality, mood and wellbeing, in a sample of university students” - line 100-111), and a second objective (“investigate the overall perceived restorative potential of both the nature and urban walk” – line 112-113); although, research hypothesis are not clearly defined in the introduction – the hypothesis are only referred, dispersed, in the methods chapter.
- Considered as a last aim, the authors “explored the individuals’ psychological experiences during the walk in the natural or urban environment, in efforts to provide insight into the underpinnings of the nature wellbeing relationship and enhance our understanding of the process of walking in a natural or urban environment” – lines 114-116.
About the Significance:
- The study was correctly designed as a randomised controlled study, however, the size of the sample should be considered too small for a quantitative comparative study. Therefore, the significance of the results is very low (however the results are appropriately interpreted), and, consequently, the conclusions, which are apparently justified and supported by the results, have very little relevance. At the article abstract and title this preliminary nature of the results is clearly identified (pilot study), also referred in discussion, although this so important aspect was totally loss in the conclusions.
- This pilot randomised controlled study is nested with qualitative interviews. The size of the sample, for a qualitative analysis, could be considered not high but appropriated to provide insights about nature-wellbeing relationships and enhance understanding of the process of walking outdoor. In this sense, the qualitative results provide an advance in current knowledge. But the quantitative results did not.
About the Quality of Presentation:
- The article is, generally, well organized and written in an appropriate way. There are a few exceptions, such as: description of the sample should be part of the methods (not results).
- Quantitative data and their analyses is appropriately presented, although the small size of the sample.
- The content analysis (qualitative analysis) needs more detailed explanation: the identification of the main themes considered, and how their relative importance on the participant’s discourses was assessed.
About the Scientific Soundness:
- The study was initially correctly designed. Due to the COVID-19 restrictions data collection was interrupted. Consequently, when referring the quantitative results, the very low size of the sample made it not technically soundness. The analysis was performed with the appropriated statistical tests but the size of the sample did not follow high technical standards.
- It is referred (lines 232-236) that the “sample size estimation prior to the study … indicated that a sample of 120 participants should be recruited in order to observe a small effect size …. Unfortunately, data collection was interrupted due to the pandemic restrictions; as a consequence, the analyses presented in this study are to be considered preliminary.” Thus, 13 participants instead of about 120 seems to be such a huge sample gap. Clearly, the quantitative data should not be considered robust enough to draw conclusions.
- Overall the methods are described with sufficient details to allow another researcher to reproduce the results, expect the qualitative thematic analysis, as already referred above.
Even so, the “preliminary” conclusions present in this article should be considered as having interesting for the readership of the Journal, and probably will attract a wide readership, specially the results and conclusions from the qualitative analysis,i.e., the insights about nature-wellbeing relationships provided and reflexions about the process of walking in naturalised settings.
Thus, and because qualitative results provide some advance towards the current knowledge on the restorative effects of nature and about the potential vitalizing effects of connecting with nature during walks, in my opinion there is an overall benefit to publishing your work as an exploratory study focused on the qualitative analysis. The quantitative results (test of hypothesis for compare urban versus naturalised walk on individuals wellbeing, vitality and mood) should be used only as a complement of the qualitative results.
According to this, in my opinion, the article will have to undergo a major revision process, in order to focus on the qualitative results.
Author Response
Response to Reviewer 2 comments
Point 1: About the Originality/Novelty of the article:
- The empirical/experimental study of the restorative effects of nature, the potential vitalizing effects of connecting with nature, should be considered as a very relevant subject, with growing importance in our modern “urban” societies. The development of such a study in a higher education setting sounds also relevant, because students face high levels of stress and anxiety. The subject sounds even more relevant at pandemic restrictions context.
- It is stated at the abstract that (lines 13-14) they aimed to “compared the effectiveness of a brief walk either in nature or in an urban environment, on subjective vitality, wellbeing, mood and perceived restorative potential” in university students, and, in the introduction, they identified a primary objective (“compare the effectiveness of a 20-25 minute walk in either a natural environment or in an urban environment with heavy traffic on subjective vitality, mood and wellbeing, in a sample of university students” - line 100-111), and a second objective (“investigate the overall perceived restorative potential of both the nature and urban walk” – line 112-113); although, research hypothesis are not clearly defined in the introduction – the hypothesis are only referred, dispersed, in the methods chapter.
- Considered as a last aim, the authors “explored the individuals’ psychological experiences during the walk in the natural or urban environment, in efforts to provide insight into the underpinnings of the nature wellbeing relationship and enhance our understanding of the process of walking in a natural or urban environment” – lines 114-116.
Response 1: We would like to extend our sincere thanks to you for taking the time to read and review our manuscript. We are very grateful to you for providing your expertise and insights regarding our research study. As you have suggested, we have changed the paper to present exploratory findings from the study focused on our qualitative analysis. Our quantitative findings are used only to complement/support the qualitative analysis. The main objectives of our study have been altered. See abstract lines 10-24 and study objectives lines 166-184 (Introduction). Additionally, we have clearly outlined our research hypotheses in the introduction. See lines 172-177. Changes have also been made to the Methods, Results and Discussion section to focus mainly on the qualitative results.
Point 2: About the Significance:
- The study was correctly designed as a randomised controlled study, however, the size of the sample should be considered too small for a quantitative comparative study. Therefore, the significance of the results is very low (however the results are appropriately interpreted), and, consequently, the conclusions, which are apparently justified and supported by the results, have very little relevance. At the article abstract and title this preliminary nature of the results is clearly identified (pilot study), also referred in discussion, although this so important aspect was totally loss in the conclusions.
- This pilot randomised controlled study is nested with qualitative interviews. The size of the sample, for a qualitative analysis, could be considered not high but appropriated to provide insights about nature-wellbeing relationships and enhance understanding of the process of walking outdoor. In this sense, the qualitative results provide an advance in current knowledge. But the quantitative results did not.
Response 2: Thank you for these very valid points. As aforementioned, based on your suggestion we have now altered the methodological focus of our study. We acknowledge the fact that our sample size is small for a quantitative comparative study and we feel that the revisions made address both this aspect and the preliminary nature of our results. See introduction lines 161-184, study design lines 186-193, lines 435-441 and 727-745 (Results section), and lines 746-770, 824-849 (Discussion Section). Additionally, we have now clearly identified the preliminary nature of our results in the conclusion section. See lines 1094-1132.
We agree that the size of the sample, for a qualitative analysis, is not very high but appropriate to provide insights about nature-wellbeing relationship. We have highlighted and justified this in our limitations section. See lines 999-1004 (Discussion).
Point 3: About the Quality of Presentation:
- The article is, generally, well organized and written in an appropriate way. There are a few exceptions, such as: description of the sample should be part of the methods (not results).
- Quantitative data and their analyses is appropriately presented, although the small size of the sample.
- The content analysis (qualitative analysis) needs more detailed explanation: the identification of the main themes considered, and how their relative importance on the participant’s discourses was assessed.
Response 3: The sample characteristics have now been moved from the results section to the methods. See lines 235-252. We have provided a detailed explanation of our approach taken with the qualitative analysis. Whereby, we justify our choice of thematic analysis and highlight the frameworks and guidelines we followed while conducting the analysis. See lines 366-398 (Data Analysis).
Point 4: About the Scientific Soundness:
- The study was initially correctly designed. Due to the COVID-19 restrictions data collection was interrupted. Consequently, when referring the quantitative results, the very low size of the sample made it not technically soundness. The analysis was performed with the appropriated statistical tests but the size of the sample did not follow high technical standards.
- It is referred (lines 232-236) that the “sample size estimation prior to the study … indicated that a sample of 120 participants should be recruited in order to observe a small effect size …. Unfortunately, data collection was interrupted due to the pandemic restrictions; as a consequence, the analyses presented in this study are to be considered preliminary.” Thus, 13 participants instead of about 120 seems to be such a huge sample gap. Clearly, the quantitative data should not be considered robust enough to draw conclusions.
- Overall the methods are described with sufficient details to allow another researcher to reproduce the results, expect the qualitative thematic analysis, as already referred above. Even so, the “preliminary” conclusions present in this article should be considered as having interesting for the readership of the Journal, and probably will attract a wide readership, specially the results and conclusions from the qualitative analysis, i.e., the insights about nature-wellbeing relationships provided and reflections about the process of walking in naturalised settings.
- Thus, and because qualitative results provide some advance towards the current knowledge on the restorative effects of nature and about the potential vitalizing effects of connecting with nature during walks, in my opinion there is an overall benefit to publishing your work as an exploratory study focused on the qualitative analysis. The quantitative results (test of hypothesis for compare urban versus naturalised walk on individuals wellbeing, vitality and mood) should be used only as a complement of the qualitative results.
According to this, in my opinion, the article will have to undergo a major revision process, in order to focus on the qualitative results.
Response 4: We agree with the Reviewer that the sample size is small and that a larger sample size would enhance the robustness of our analysis. However, we are not in a position to continue data collection for the foreseeable future due to the pandemic restrictions. We hope that the revisions made to the paper, presenting the qualitative results as the main point of the study and the quantitative results as preliminary, have enhanced the quality of our manuscript. We appreciate your acknowledgment that there is an overall benefit to publishing our work due to the fact that the qualitative results provide some advance towards the current knowledge on the restorative effects of nature and about the potential vitalizing effects of connecting with nature.
Round 2
Reviewer 2 Report
Dear Authors
Thank you for accepting and following the reviewrs suggestions.
I agree there is an overall benefit to publishing your work as an exploratory study because the results provide some advance towards the current knowledge on the restorative effects of nature and about the potential vitalizing effects of connecting with nature.
Although, there is a clearly need for further data collection; even if that is not possible in the current circumstances, I hope that you will make it possible in the near future, the further development of this study.
Author Response
Response 1: We would like to extend our sincere thanks to you for taking the time to read and review our revised manuscript. We are thankful to you for acknowledging the benefit of publishing our work and we are very grateful to you for providing your expertise and insight regarding our research study. We are in agreement that the quantitative element of our study requires further data collection to warrant publication. Thus, we have removed all inferential tests from the paper which is now presented as a qualitative study. We only present quantitative findings to explain the population characteristics, walking preferences and the restorative potential of each walk. We have altered the title of our study: “Natural or Urban Campus Walks and Vitality in University Students: Exploratory qualitative findings from a pilot randomised controlled study.” (See lines 1- 4). Additionally, we have changed the abstract (9- 22), study objectives (185- 196), study design (199-208) and removed all quantitative findings from the study: Measures (275- 300), Procedure (301- 318), Data Analysis (448- 449; 499-504), Results (505- 799), Discussion (800- 1109). Lastly, we have rearranged the qualitative findings so that they are more clearly organised. To address we have split the qualitative findings into three distinct sub-sections with each theme as the heading. We have also spaced out the participants' comments and the analysis so that they are easier to read and understand. See Results (lines 505- 799).